# Climate influence on the early human occupation of South America during the late Pleistocene

L. Becerra-Valdivia [1,2] ✉

The settlement of South America marks one of the final steps in human expansion. This study examines the impact of climate change on this process, focusing on two millennial-scale climatic phases—the Antarctic Cold Reversal and Younger Dryas. Using Bayesian chronological modelling, a cultural timeline was constructed from approximately 150 archaeological sites and 1700 dates, and compared against paleoclimatic records. Findings suggested that human activity likely began in regions most affected by the Antarctic Cold Reversal, specifically in southernmost and high-altitude areas. Together with estimates indicating that the onset of megafaunal exploitation and bifacial point technology occurred before or during the Antarctic Cold Reversal, results suggested that cold conditions did not likely hinder human settlement. Key factors likely included accumulated cultural adaptation and relatively milder climatic changes in the Southern Hemisphere. More widespread occupation likely occurred during or, more likely, after the Younger Dryas as conditions stabilised. Results highlighted the western Andes as a crucial dispersal route and questioned the role of humans and climatic shifts on megafaunal extinctions. An analysis of the compiled archaeo-chronometric dataset revealed significant under-representation and reporting gaps, highlighting the need for expanded research and rigorous documentation to improve the reliability of the cultural timeline.

The early settlement of South America around the last Ice Age (ending ~12 thousand years ago or 'ka') is one of the final stages in human expansion across the planet. Humans first entered the region from North America, where widespread settlement likely began during a period of abrupt climatic warming in the Bølling-Allerød (BA) chronozone (~14.6–12.9 ka before AD 2000)[1,2]. Although North American findings provide valuable insights into the early settlement of South America, this process should be evaluated independently from a climatic perspective due to significant interhemispheric differences, driven by the bipolar seesaw mechanism, which result in contrasting climatic trends[3,4]. Crucially, whilst the Northern Hemisphere (NH) experienced abrupt warming during the BA, followed by cooling and glacial readvance through the Younger Dryas (YD; starting at ~12.9 ka before AD 2000)[1], the Southern

Hemisphere (SH) generally saw inverse climatic conditions; cooling caused by the Antarctic Cold Reversal (ACR; ~14.7–13 ka)[5,6], followed by increasing temperatures, precipitation and deglaciation in the YD[7–12]. Despite these contrasts, the impact of hemisphere-specific climatic phases in late Pleistocene archaeology within South America remains understudied. To address this, the present study examines the role of the ACR and YD in the early settlement of South America by constructing a cultural (i.e., human-occupation) timeline and contextualising the findings within paleoclimatic evidence. This involved:

i) compiling previously published data from approximately 150 archaeological sites and classifying them based on key attributes (e.g., geographic province, lithic technology, altitude and megafauna kill/scavenging evidence; Fig. 1)

[1]Department of Anthropology and Archaeology, University of Bristol, Bristol, UK. [2]Linacre College, University of Oxford, Oxford, UK.
✉e-mail: lorena.becerra-valdivia@bristol.ac.uk

ii) integrating and evaluating the archaeo-chronometric data at a site level using Bayesian chronological modelling;

iii) identifying large-scale spatiotemporal patterns by correlating multiple sites through the integration of site-specific outputs—represented as probability density functions (PDFs)—into models informed by the shared classifications (e.g., geographic province; Fig. 2); and

v) comparing the results with the timing of both ACR and YD, alongside region-specific palaeoclimate data (see Methods).

Given that the cultural timeline is mainly underpinned by radiocarbon dating, this study emphasised the compilation of [14]C quality-control data to assess measurement reliability. Further, because the ACR and YD are specific to the Southern Hemisphere as relatively cold

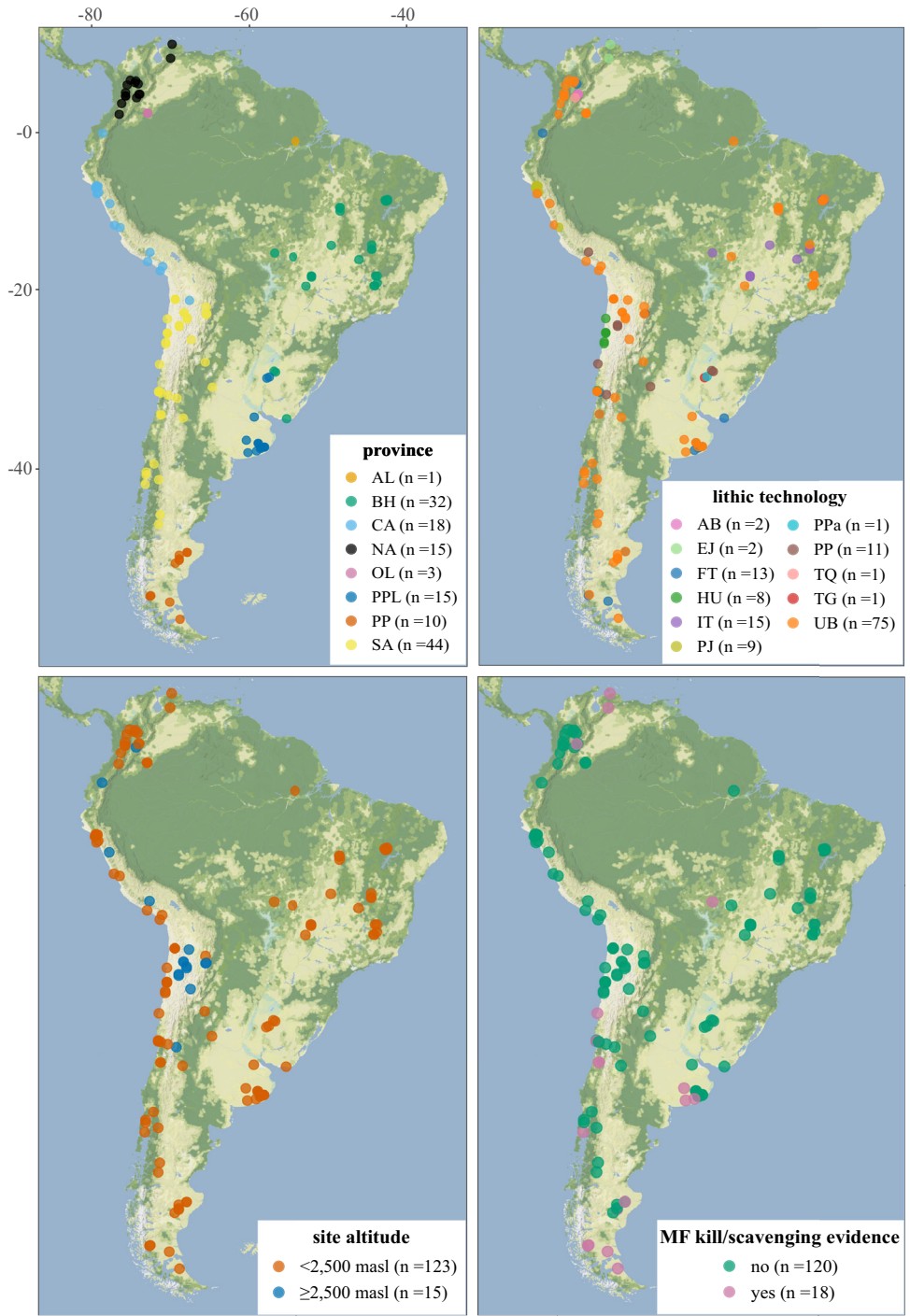

**Fig. 1 |** Spatial distribution of archaeological sites included in the Bayesian chronological modelling according to geographic province, lithic technology (tradition or category), altitude and evidence for megafauna (MF) killing/scavenging by humans. Although a single lithic tradition/category is assigned to each site, some contain more than one (e.g., Pay Paso 1, Uruguay; see Supplementary Note 2) so this map should be seen as illustrative. The modelling work (e.g., Fig. 2) takes different cultural components and specific lithic traditions/categories into account. AL = Amazonian Lowlands (dark orange); BH = Brazilian Highlands (teal green); CA = Central Andes (sky blue); NA = Northern Andes (black); OL = Orinoco Lowlands (magenta pink); PPL = Paraguay-Paraná Lowlands (deep blue); PP = Patagonian Plateau (burnt orange); SA = Southern Andes (pale yellow); AB = Abriense (pink); EJ = El Jobo (light green); FT = Fishtail (blue); HU = Huentelauquén (green); IT = Itaparica (purple); PJ = Paiján (olive green); PP = projectile point (brown); PPa = Pay Paso (cyan); TG = Tigre (red); TQ = Tequendamiense (light red); UB = uniface/biface (orange). <2,5000 masl = orange. ≥2,5000 masl = blue. MF kill/scavenge evidence no = green, yes = pink. Source data can be found in Supplementary Data 1.

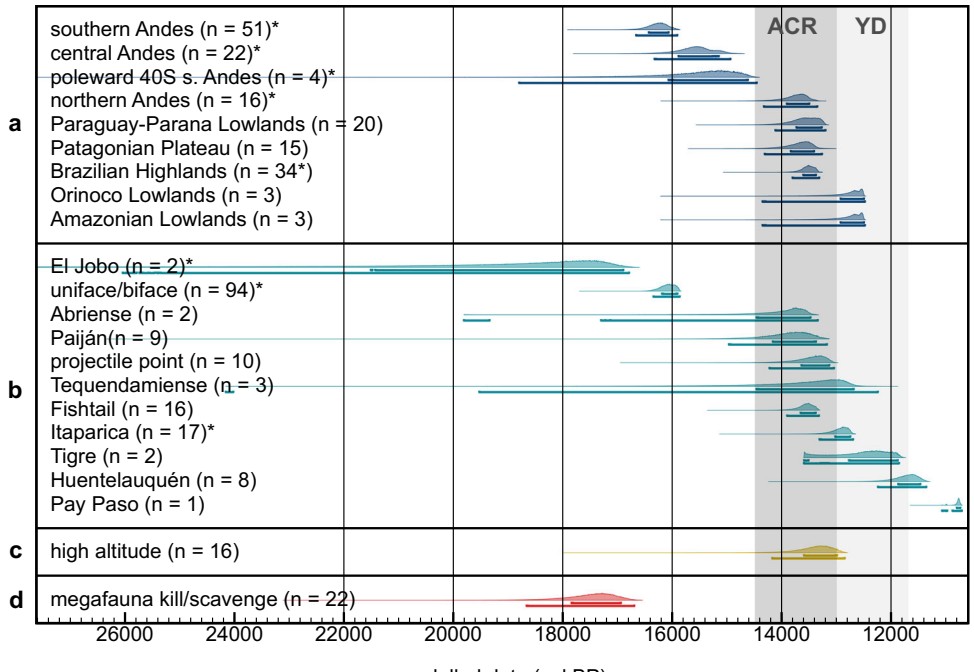

**Fig. 2 | Bayesian chronological modelling results.** Start estimates for ACR/YD-aged cultural components are arranged according to (**a**) province (blue) and (**b**) lithic tradition/category (teal), as well as (**c**) high-altitude (≥2,500 m.a.s.l.) occupation (yellow) and sites with (**d**) evidence of megafauna killing/scavenging (red). The sample size is next to each distribution name, noting the number of cultural components that were included to create each model (see Supplementary Fig. 1-8, 12-21, 22, 23). Categories marked with an asterisk (*) represent provinces and lithic technologies with limited cultural evidence—both in sample size and scholarly agreement—that predate the ACR and do not temporally extend into this period (see Supplementary Fig. 10). Brackets beneath each distribution denote 68.2% and 95.4% CI. The timing of the ACR and YD are denoted in dark and light grey bands, accordingly. Source data can be found within the Source Data file. OxCal code is in Supplementary Note 3.

and warm events, respectively, results and implications mainly focus on SH South America. It is important to note, however, that there is considerable climatic variability within this region. The ACR, for instance, displays a south-to-north gradient, with its effects most pronounced south of 40° S[5] and a weakened signal at lower latitudes, poleward of ~20° S[10,13,14]. As a result, cooling had a significant impact in southernmost provinces, such as the Patagonian Plateau and the southern Andes, where glacier readvance is documented[3,12,15–17] and was also likely facilitated by intensified Southwesterly Winds (SWW)[18–22] (which bring moisture from the Pacific Ocean). Glacier readvance here is not limited to the ACR, however, also occurring during the YD[23,24]—a period of heightened SWW variability[25]. Northward, both ACR and YD in central Andes were marked by wetter conditions attributed to the Central Andean Pluvial Event and an enhanced South American Summer Monsoon System, which led to the expansion of wetlands and lakes[26,27]. Here, as in northern Andean areas, glacier readvance is linked to regional increases in precipitation[10,13,14,28]. In the Paraná-Paraguay Lowlands, Brazilian Highlands, and Amazonian Lowlands, there is evidence for reduced precipitation during the ACR, and an increase during the YD[9,11,29–31]. Conversely, northernmost South America, including the northern Andes and Orinoco Lowlands, reflects a Northern Hemisphere climatic signal, showing warmer, wetter conditions during the BA/ACR period[11,32,33], and colder, drier conditions during the YD or immediately after[32,33].

In this work, the Antarctic Cold Reversal does not appear to be detrimental to the late Pleistocene settlement of South America, as human activity begins or persists in regions most affected by cold conditions (southernmost and high-altitude). This pattern is likely due to accumulated cultural adaptation and relatively milder climatic shifts in the Southern Hemisphere. A west-to-east settlement trend in the south reinforces the western Andes as a key dispersal route. Widespread occupation of the continent occurs during or, more likely, after

the Younger Dryas as conditions stabilise, suggesting a lag in human adaptation to warming compared to North America. Regarding megafaunal extinctions, no causal link to human activity or climatic shifts could be determined. Finally, the archaeo-chronometric dataset reveals significant underrepresentation and reporting gaps, which may affect the construction of reliable cultural timelines.

## Results
### Bayesian chronological modelling

Given the south-to-north gradient in the strength of the ACR[5], start boundaries for ACR/YD-aged cultural components according to eight geographic provinces (see Methods for their definition) were produced (Fig. 2a; Supplementary Fig. 1-8). Results, including overlap testing in Supplementary Fig. 9, show that cultural activity in the southernmost portion of the continent—as represented by southern Andes [16,660–15,905 cal BP ('cal BP' = calibrated years before AD 1950)] and the Patagonian Plateau (14310–13265 cal BP)—likely began before or during the ACR (Supplementary Fig. 1, 6, 9). Distributions from northern provinces also denote an ACR start (apart from later Orinoco and Amazonian Lowlands estimates). Note that in certain provinces, including the southern Andes, there are claims of pre-ACR cultural evidence, representing a comparatively limited number of sites whose antiquity is debated (Fig. 2 asterisks; Supplementary Fig. 10, with further details in Supplementary Note 2). Considering the latitudinal span of the southern Andes, the model was repeated for this province using only cultural components poleward of 40° S to estimate the start of human activity for the southernmost portion and validate the findings. This estimate, at 18795-14460 (95.4% CI) or 16065–14620 cal BP (68.3% CI), shows the same pattern to hold (pre- or ACR start; Supplementary Fig. 3). The spread and antiquity of this age range, as with other PDFs in this study with the same characteristics, is likely due to increased uncertainty relating to limited sample size (in

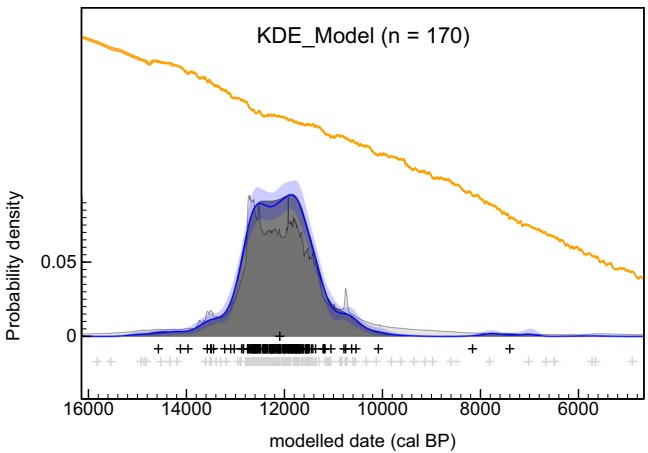

**Fig. 3** | Summarised distribution of archaeo-chronometric data for the ACR/YD-aged cultural components analysed (*n* = 170), as represented by `R_Date` and `Date` (see Methods). The `KDE_Model` distribution is at the forefront, with a noisier `Sum` distribution behind in lighter grey. The rug plots show the median values of the date ranges (grey crosses) and the medians of the marginal posterior distributions for each event (black crosses). The blue line and lighter band represent the mean ±1σ for snapshots of the KDE distribution produced during the MCMC process. The IntCal20 curve[35] ±1σ is shown in orange for reference. Within the distribution produced, data are concentrated around a single peak between -12750 and 11500 cal BP (mean = 12050 cal BP; median = 12095 cal BP). Source data can be found in Supplementary Data 1. OxCal code is in Supplementary Notes 3.

this case, *n* = 4) and lack of chronological constraint from more basal levels (or other prior information). The data also show a west-to-east trend in southern South America, with an earlier start of occupation west of the Andes compared to eastern regions (Patagonian Plateau and Paraguay-Parana Lowlands)—the distributions for the poleward 40° S southern Andes and Patagonian Plateau do not overlap (Supplementary Fig. 11; see Supplementary Data 2 for GIF).

Start estimates for eleven lithic technologies, including formal traditions (e.g., Tigre; see Supplementary Note 1 for descriptions) and categories (e.g., uniface/biface) are plotted in Fig. 2b (see Supplementary Fig. 12–21). Results show that five of the eleven traditions/categories likely began during the ACR, with three starting during or immediately after the YD. Considering potential pre-ACR evidence[34], results of this analysis suggest that bifacial point technology in South America likely began by or around the ACR—with start estimates for Paiján at 14,960–13,175 cal BP and projectile point technology at 14,220–13,045 cal BP (Supplementary Fig. 13-14). There is no clear spatial or otherwise technological, i.e., uni- vs. bi-face, pattern to the data (see Supplementary Data 3 for GIF).

Considering the observed impact of ACR-induced cooling in high-altitude Andes[10,13,14,22], cultural components from archaeological sites located ≥2500 metres above sea level (m.a.s.l.) were also analysed. Results show that the cultural occupation of high-altitude Andes likely commenced during the ACR, at 13750–12830 cal BP (or 13385–12935 cal BP at 68.3% CI; Fig. 2c, Supplementary Fig. 22), with no clear spatial trend (see Supplementary Data 4 for GIF).

Given archaeological information for each site (see Supplementary Note 2 for reports), cultural components with evidence of megafauna killing/scavenging (*n* = 21) were introduced into a single-phase model to estimate the start of this activity (Supplementary Fig. 23). Results show that the killing/scavenging of megafauna by humans likely began before the ACR, at 18,660–16,695 cal BP (Fig. 2d). Apart for the location of the two pre-ACR components located equatorward of 40° S (Santa Elina and Taima-Taima), there is no clear spatio-temporal pattern observed thereafter (see Supplementary Data 5 for GIF).

Finally, a density plot was produced to view the temporal distribution of cultural components that overlap the ACR-YD period (Fig. 3). This shows a peak between -12750 and 11500 cal BP, and a relatively unimodal distribution (mean = 12050 cal BP; median = 12095 cal BP).

### Database outline
The database created includes approximately 150 archaeological sites and over 1700 ages—the majority of which are radiocarbon measurements (>90%)—that are pertinent to the period/region studied. Certain regions, like the Guiana Highlands, Orinoco Lowlands and Amazon Lowlands, are either entirely or largely underrepresented (Supplementary Data 1; Supplementary Data 2 for GIF). In contrast, countries like Chile and Brazil have the highest proportion of measurements and archaeological sites in the continent (Supplementary Fig. 24). Regarding radiocarbon data, charcoal accounts for more than half of all measurements, but the species are mostly unreported (94%). Additionally, crucial details such as pretreatment type, species identification, and elemental and isotopic values are largely missing (Supplementary Fig. 25). The percentage of reported carbon-to-nitrogen ratios for bone collagen, for example, is only 11. However, of those reported (*n* = 28), all fall within the accepted range for bone collagen[35]. This means that either the samples are not significantly contaminated, or that the laboratories involved followed quality control criteria and no samples with values outside of the range were reported.

### Discussion
Results indicate that human activity likely began or continued in areas most affected by cold temperatures (southernmost and high-altitude), implying that the ACR did not act as a major influence on the late Pleistocene settlement of Southern Hemisphere South America. Two probable explanations can be drawn from this. First, humans were likely already present in certain regions prior to the ACR[34,36–44] (see Supplementary Fig. 10), with direct evidence of megafauna exploitation and bifacial point technology starting before or during this time (Fig. 2b, c). Secondly, changes in temperature during the ACR/BA were likely less drastic than in the Northern Hemisphere[45] (see Supplementary Fig. 26-27), and glacier re-advance in the Andes and Patagonia was not widespread or synchronous[13–16,24]. This suggests that accumulated cultural adaptation and comparatively milder climatic changes may have played an important role in the survival of human groups during the ACR. The concept of accumulated cultural adaptation—drawn from the work of Richerson and Boyd[46,47]—refers to the cumulative, intergenerational transmission of knowledge and technologies (i.e., culture) that likely provided ancient humans with an adaptive advantage. The west (early) to east (later) trend observed in the southernmost region might also be explained by pre-ACR human activity in the Andes[34,36–40], although climatic factors are worth mentioning. In mid- to high-latitudes, there is a west-east antiphase with wet (west) and dry (east) conditions on either side of the Andes caused by the Southwesterly Winds and the rain shadow effect[48]. Given that humans were present in lower latitudes along the Andes where this antiphase pattern is reversed, however, a purely climatic explanation is unlikely, despite the differing environments. Importantly, this west-to-east pattern highlights the Andes (rather, the Pacific Coast) as an important route for the initial dispersals into South America. This is consistent with numerical models[49,50] and genetic evidence showing the earlier settlement of the Andes/Pacific versus eastern regions[51–53]. Regarding high-altitude occupation, Andean sites in northern Chile and southern Peru probably benefited from wetter conditions brought about by the Central Andean Pluvial Event.

Results pertaining to human-megafauna dynamics indicate that exploitation of the latter began considerably earlier—sometimes millennia—than last appearance dates (LADs; extracted from Prates and Perez)[54] for genera found in closely associated archaeological contexts

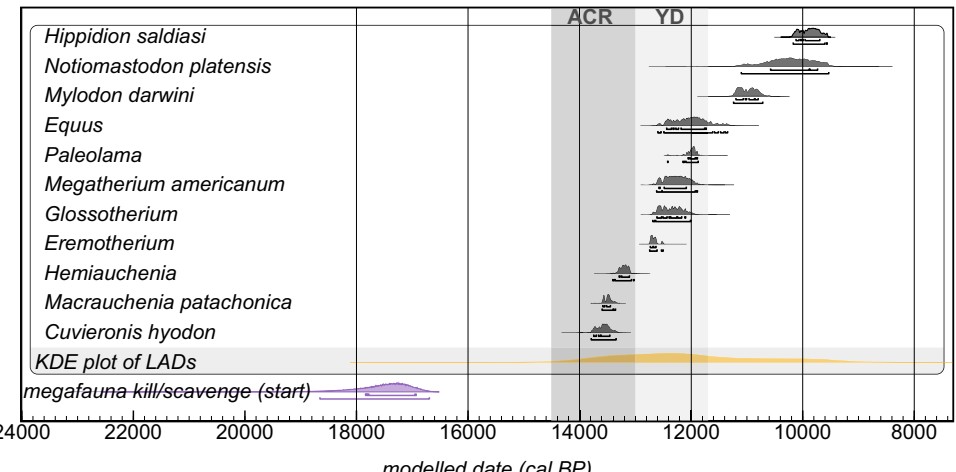

**Fig. 4** | Last appearance dates (LADs; grey) for fauna present in archaeological contexts with direct evidence of killing/scavenging in relation with the estimated start for human exploitation (purple; 18660-16695 cal BP). Brackets beneath each distribution denote 68.2% and 95.4% CI. The timing of the ACR and YD are denoted in dark and light grey bands, accordingly. A KDE plot of the LADs, which shows the temporal distribution, appears in orange. This is multimodal with a mean = 12060 cal BP and a median = 12240 cal BP. Source data can be found in the Source Data file and in Prates and Perez[54] (for the LADs). OxCal code is in Supplementary Note 3.

(Fig. 4; see Supplementary Note 2 for site reports). This temporal gap, combined with the inherent limitation of LADs, which record the death of a single individual rather than the point of effective species extinction or decline (the latter likely more informative), precludes establishing a definitive causal link between human exploitation and megafaunal extinction. Whilst quantitative analysis between the results of this study and LADs is possible (as relationships between any distributions can be tested), such comparisons are not advisable as they involve fundamentally different temporal markers—modelled start estimates for human activity vs. unmodelled, *terminus post quem* ages for faunal extinction. Further, there are currently no genetics-based demographic reconstructions, like those based on data from extant South American species[55–58] or extinct fauna elsewhere[59,60], to elucidate the matter. Similarly, a clear link between these extinctions and climatic changes associated with the ACR/YD remains elusive. These findings contrast with analyses that attribute megafaunal extinctions in South America primarily to human activity or climate change as the dominant factor[61–65].

By the Younger Dryas onset, results suggest that human occupation had likely begun in all represented provinces (Fig. 2a; see Supplementary Data 2 for GIF), with a coeval increase in archaeo-chronometric data (Fig. 3). This increase should not be immediately viewed as a correlation between the start of warmer conditions with population growth, as the frequency of radiocarbon dates may not directly correspond to population size (see Methods and reference)[66]. Genetic data points to a rapid expansion into South America between 15,000 and 13,000 years ago, marked by the widespread presence of specific haplogroups and closely spaced divergence events[67]. Yet, there is limited focus on diachronic patterns of population size during the ACR/YD, with evidence suggesting that increases began only after this period[68]. Therefore, the cultural occupation of South America can be confidently regarded widespread as warming conditions stabilised post-YD, with humans in previously uninhabited regions engaging in specialised activities, such as plant domestication[69,70]. This pattern indicates a potential lag in human adaptation to warming compared to North America, where evidence places the emergence of distinct technological traditions during[71], or in response to, warming[2]. This difference may be explained by the abruptness of the Bølling-Allerød[72], cultural factors, or a combination of both. Regardless of the speed at which humans responded to climatic changes, it appears that whilst cold conditions during the ACR were not entirely detrimental, warming facilitated expansion. The latter is also observed outside of the

Americas in Northwestern Europe, for example, which was largely repopulated following the Last Glacial Maximum by the Bølling-Allerød[73].

Finally, the collated results underscore the underrepresentation of certain provinces and countries in both archaeological and chronometric records, as well as the inadequate reporting of chronometric data. This underrepresentation likely results from a combination of taphonomic processes, research and funding biases, and poor sample preservation for radiocarbon dating in hot, humid regions such as the Guiana Highlands and the Orinoco and Amazon Lowlands. Inadequate reporting of chronometric data further complicates the assessment of radiocarbon measurement reliability. Critical information, such as pretreatment methods, species identification, and elemental and isotopic values, is essential for radiocarbon specialists to evaluate factors like sample preservation, contamination, and inbuilt age (including non-atmospheric reservoir offsets)[35,74]. Given that the period studied is relatively recent in radiocarbon dating terms, issues such as modern carbon contamination are less problematic compared to older periods, like the European Middle-to-Upper Palaeolithic transition, where a significant proportion of dates are likely too young[75]. Another issue linked to research and funding biases is the limited chronometric data available for stratified sequences in South America; often, a single cultural layer may be represented by zero, one, or just a few measurements. For Bayesian chronological modelling, this results in temporal estimates that are likely accurate but lacking in precision, potentially leading to artefacts that could be misinterpreted. To address these challenges, increasing the frequency of dating in stratified sequences, expanding research in underrepresented provinces (considering indigenous communities and ethical implications), and ensuring comprehensive reporting of chronometric data are crucial. These steps would significantly enhance the accuracy and precision of the cultural timeline presented here and future efforts.

## Methods

There are no ethical regulations that apply to this work and, as such, no permissions were required.

Archaeo-chronometric data for sites containing ACR/YD cultural features were compiled in Supplementary Data 1, without exclusion (whether based on precision, rejection by the archaeologist or otherwise). This includes ages outside the ACR-YD (both older and younger), to ensure comprehensive reporting and the modelling of archaeological sequences in their entirety (more details below). The

bibliographic reference for each data point is that from which the information was obtained. Generally, this corresponds to the original source. For radiocarbon calibration, if no material type was reported, the sample was assumed to be from an atmospheric environment and calibrated accordingly using IntCal20[76], SHCal20[77] or a mixed curve approach (see below). Marine radiocarbon measurements, which are affected by the marine reservoir effect[78], were calibrated using the Marine20 calibration curve[79] with a ΔR correction to account for regional variations from the global average. Specific ΔR values used are noted in Supplementary Note 2 (the site report and/or the OxCal code). Due to the likely influence of NH air masses in north-central SA, i.e., parts of the Amazonian Lowlands and Brazilian Highlands, the `Mix_Curve` command with a uniform distribution was utilised for a mixed hemispheric curve[80]. Spatial guidelines presented by Marsh et al.[81] were followed (see their Fig. 4), only impacting a comparatively limited number of sites ($n = 10$) between ~25 S° and 15 N°.

The cultural timeline was created using Bayesian chronological modelling in the OxCal platform (version 4.4)[82], which uses Chronological Query Language (CQL; here noted in Courier font for accessibility). Briefly, Bayesian chronological modelling is a statistical method that analyses chronometric data (standardised likelihoods) in the context of existing knowledge (priors) to produce results (posterior probabilities). Priors can include absolute or relative information, including archaeological stratigraphy[83]. The results are expressed as probability density functions using Markov chain Monte Carlo (MCMC) sampling, which combines the Metropolis-Hastings algorithm and Gibbs sampling[84,85]. Default OxCal parameters include an initial 30,000 iterations with convergence checked every 3,000 (a 'pass'), with the pass interval increased by a factor of two until convergence is satisfactory (>95%)[86].

In archaeological applications, Bayesian chronological modelling functions by splitting time into units, treating archaeological components as groups defined by double (start and end) boundaries. A group in OxCal can be represented by a `Phase` (for unordered events) within a `Sequence` (for ordered events) bounded by two `Boundary` commands. The latter serve to define the distribution of the events according to pairing variations. A `Boundary-Boundary` pair, for instance, defines a uniform phase that assumes all events are equally likely to occur within the boundaries. Apart from defining groups and event distributions, boundaries also provide estimates for the start or end of otherwise undated cultural events. This is particularly informative in archaeological applications given that the earliest cultural date for a site or region likely reflects a minimum age rather than the precise moment of human presence or arrival. Other informative MCMC and non-MCMC functions include `Date`, which produces an age estimate for dated or undated cultural events; `R_Combine`, which checks the consistency of multiple ages[87] belonging to the same (or by a factor of 10 relative to the measurement error) moment in time, e.g., duplicate ages on a single sample; and `KDE_Plot` or `KDE_Model`, which estimate the underlying distribution of multiple dates using kernel density estimation (KDE)[88,89,90]. The latter, `KDE_Model`, is an MCMC-based method designed for analysing sets of related dates. It utilises a normal kernel and assumes a normal distribution, with bandwidth estimated according to Silverman's rule[91]. To address the issue of over-smoothing in multi-modal distributions, a shaping parameter is applied to the bandwidth, and events within a grouping are treated as dependent. In OxCal, the default kernel and factor parameters applied relative to Silverman's rule are N(0,1) and U(0,1), respectively. Finally, outlier analysis (OA) addresses the assumption that all dates within a group are accurate. By assessing how well chronometric data align with the prior framework, OA objectively identifies outliers and automatically downweights them based on the degree of offset. In OxCal, the distribution, magnitude, and prior probability of outliers can be defined using a number of preset options, including `General`, `SSimple`, and `Charcoal`[92], using `Outlier_Model()` and `Outlier()`

commands. The `General` outlier model, for example, uses a Student's t-distribution with 5 degrees of freedom to account for extreme outliers, with a scale of anywhere between $10^0$ to $10^4$ years.

Bayesian chronological modelling of archaeo-chronometric data fulfills three key functions:

i) At its core, it allows for the integration of archaeological and chronometric data. This is simple, but essential; a radiocarbon measurement reflects the moment an organism ceased exchanging carbon with its reservoir, which may not correspond to the timing of a cultural event. For instance, dating a carved bone tool provides the age of the animal's death, not necessarily when the bone was carved or deposited in a burial. The temporal gap between these events can vary from minimal (within calibration uncertainty) to substantial (if older fossil bone is used), but it is crucial to consider. Radiocarbon measurements, therefore, must be interpreted within their broader archaeological context to ensure meaningful interpretation.

ii) Bayesian chronological modelling enables the statistical, highly informative assessment of archaeological sequences. Outlier analysis and its down-weighting function, in particular, provides a transparent, objective approach—eliminating the need to manually exclude dates—whilst offering valuable insights. For instance, outlying dates within a sequence often reveal issues such as vertical mixing, taphonomic processes, inbuilt age, and/or incomplete decontamination. Without proper analysis, these outliers may go unnoticed by researchers, potentially skewing numerical studies if fully incorporated. Bayesian chronological modelling is also highly flexible, allowing for the integration of imprecise yet accurate ages and geological measurements to constrain cultural evidence—data that might otherwise be manually excluded.

iii) In tandem with the second function, Bayesian chronological modelling facilitates the correlation of multiple, statistically evaluated archaeological sequences at a large scale. A notable example is the study by Higham et al.[93], which quantified the overlap between modern humans and Neanderthals in Eurasia, by establishing the start and end of lithic technologies associated with each group across 40 archaeological sites.

These functions give Bayesian chronological modelling a distinct advantage over frequentist approaches (e.g., [14]C-dates-as-data analyses), which decontextualise measurements from the archaeological record, weighing them equally in numerical analyses, and often rely on manual exclusions that lack the objectiveness and informativeness offered by the Bayesian approach. Furthermore, from a radiocarbon dating perspective, factors such as sample decontamination and failure rates directly impact both the reliability of chronometric datasets and their representativeness of the archaeological record—particularly if measurement frequency is used to extrapolate population density, i.e., demography, as with many [14]C-dates-as-data studies. These issues, if unidentified, risk distorting distributions and highlight the need for critical, context-aware assessments of chronometric data.

In this study, Bayesian model construction for archaeological sites (see Supplementary Note 2 for site reports) followed a set of rules:

1. If the site contained multiple stratigraphic levels and measurements, a multi-phase [uniform (e.g., `Boundary-Boundary`)] model was created based on stratigraphic information. All archaeo-chronometric data were incorporated, modeling each site in its entirety. This involves the inclusion of layers or phases beyond the ACR-YD period, as they serve to inform the model (i.e., ages in a preceding layer constrain the following layer). `General` and `SSimple` (only for replicate measurements) outlier models were used throughout, with default parameters.

2. If ≥2 measurements, the site was represented by a uniform, single-phase (uniform) model. This is also the case for sites that lacked sufficient information, e.g., clear stratigraphic descriptions, for multi-phase modelling.

3. If represented by only one (non-sediment) measurement dating a single cultural event, no site-specific modelling was done; the single measurement was used.

Outputs (PDFs) from these approaches (see Supplementary Fig. 28) were incorporated into single-phase models to correlate multiple sites across four categories: geographic province, lithic technology, altitude, and evidence of megafauna killing/scavenging. This allowed estimation of the onset of ACR/YD-aged human activity for each category (Supplementary Fig. 1-8, 12–21, 22, 23). The distribution of ACR/YD-aged cultural components was visualized using `KDE_Model` in Fig. 3. For sites modeled under rules 1 and 2, the `Date` function was used to represent the temporal age of cultural events, whilst for single-measurement sites under rule 3, the `R_Date` function was applied.

Geographic provinces followed those in Orme[94]—Patagonian Plateau, Andes, Paraguay-Parana Lowlands, Brazilian Highlands, Amazonian Lowlands, Guiana Highlands and Orinoco Lowlands—with a further split of the Andes into northern (Venezuela, Colombia and Ecuador), central (Peru and Bolivia) and southern (Chile and Argentina) regions. Note that there are comparatively limited reports of pre-ACR evidence in South America (see asterisks in Fig. 2, Supplementary Fig. 10, and site reports within Supplementary Note 2). Therefore, the estimated start pertaining to, for example, the Brazilian Highlands, does not necessarily denote timing of *initial* human arrival. PDFs for the start of each category were statistically compared using the `Difference` command to test their overlap with ACR and YD periods, and identify spatio-temporal patterns. Sensitivity tests followed the same approach, evaluating robustness by assessing the impact of different priors on the posteriors. The ACR-YD period was temporally defined following Pedro et al.[5], who employ Antarctic ice-core records[6,8]. These, which are synchronised with Greenland datasets [the Greenland Ice Core Chronology 2005 (GICC05) timescale][95], exhibit a centennial lag in cooling response during the BA/ACR[6]. It is important to note that the GICC05 and radiocarbon timelines are not equivalent, yet they coincide within a few decades during the BA and YD[96]. To account for this and the lag, the ACR-YD period was converted to the radiocarbon timeline (cal BP or calibrated years before AD 1950) and entered into the Bayesian models as uniform distributions 14,500–13,000 cal BP [ACR; `Date(U(calBP(14500),calBP(13000)))`] and 13,000–11,700 cal BP [YD; `Date(U(calBP(13000),calBP(11700)))`].

Unless otherwise noted, all age estimates are noted at 95.4% credible/confidence intervals (CI) and rounded to 10 years (resolution for tree-ring records within the SHCal20 for the Younger Dryas is decadal)[77]. OxCal code for all models is included in Supplementary Note 2 and 3. Mathematical details for all functions/commands can be found in the OxCal manual online[97].

### Reporting summary

Further information on research design is available in the Nature Portfolio Reporting Summary linked to this article.

## Data availability

Source data for all figures can be found in the Supplementary Information file, Supplementary Data 1 and/or Source Data file. The Supplementary Information file contains brief descriptions of the formal traditions and/or categories applied to each archaeological site, as well as reports—including Bayesian chronological modelling details, figures, and OxCal code—for the latter, ordered alphabetically by country. OxCal code for Figs. 2–4 can also be found within the Supplementary Information file. The Supplementary Data 1 is a table containing all archaeo-chronometric data analysed. The Source Data folder contains the prior files needed to run Figs. 2 and 4 on OxCal (code in Supplementary Note 3). Source data are provided with this paper.

## Code availability

All code for the Bayesian chronological modelling is included in the Supplementary Information file. OxCal code is either accompanying each archaeological site report (Supplementary Note 2) or under Supplementary Note 3.

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

## Acknowledgements

The author thanks the Leverhulme Trust Fund, for funding this research through an Early Career Research Fellowship (ECF-2022-532), and Linacre College (University of Oxford; via a Bryan Warren Junior Research Fellowship), for their support. Without implying their consent to the content, I am also grateful to the following colleagues for providing feedback, expertise, and/or engaging in stimulating conversations during this work: Research Asst. Prof Bethan Linscott, Prof Thomas Higham, Dr Mariana Sontag-González, Dr Monty Ochocki, Asst. Prof Francisca Santana-Sagredo, Dr Constanza de la Fuente, Dr Rodrigo Loyola, Dr Haidee Cadd, Assoc. Prof Katerina Douka, and Prof Victoria Smith.

## Author contributions

L.B.V. collated, analysed and interpreted the data, and authored both the main text and the supplementary materials.

## Competing interests

The author declares no competing interests.
