## [Transparent Peer Review file · Nature Communications]

Climate influence on the early human occupation of South America during the late Pleistocene

Corresponding Author: Dr Lorena Becerra-Valdivia

Version 0:

Reviewer comments:

Reviewer #1

(Remarks to the Author)

The manuscript by Becerra-Valdivia, entitled "Climate influence on the early occupation of South America during the late Pleistocene," attempts to compare Bayesian chronometric models with paleoclimatic proxies (paleotemperature curves) to evaluate a probable link between early peopling dynamics and late Pleistocene climatic changes in South America. It is well-written, the methodology used is robust, and this study fits well with recent previous studies in South and North America. Despite this, I found several issues that must be addressed, and some of them call into question some of the major findings of this study. Some of my suggestions very likely imply extensive editing of the manuscript, perhaps redoing some analyses, so my suggestion is for a major revision.

I have several major and minor suggestions:

- 1) The inferred relationship between human activity and paleoclimatic change is ambiguous. A robust way to evaluate the magnitude of such an association is by using the dynamic time warping (DTW) approach, which allows us to measure the similarity (or dissimilarity) between two or more timelines. By using such an approach or a similar one, the author can demonstrate, rather than assume, an association and then defend the role of climate changes in early human peopling dynamics in South America.
- 2) The author focused on two climatic phases, ACR and YD. However, regional variations complicate extrapolations to the complete South American subcontinent. For instance, late Pleistocene climatic change in Northern South America and most of Eastern South America was influenced by seasonal variations in wind and rainfall dynamics, that is, the so-called Intertropical Convergence Zone (ICZ). Given its importance for regional human peopling dynamics, at least in Eastern and Northern South America, I recommend including additional proxies from the Cariaco Basin. The author should also consider using other regional proxies like the Sajama Ice Core from Bolivia important to the Central Andean archaeological record.
- 3) In the 14C dataset included as supplementary materials, there are a lot of 14C dates below 7000 14C BP. Were such dates used in the Bayesian modeling for the early stage of human peopling? If so, why? I understood the author assessed cultural phases by using Bayesian modeling, but it is not clear to me why such dates are included in a study focused on the late Pleistocene.
- 4) The author mentioned she used ~ 1700 14C dates, however, the criteria used to include or exclude 14C dates are unclear. For instance, a basic issue to exclude a 14C date would be its high error, but in such a dataset there are several 14C dates with a high error, especially those obtained decades ago using beta-counting techniques. The author needs to clarify the quality control procedures followed.
- 5) I suggest caution with the use of certain not-well-defined lithic techno-complexes in the South American archaeological record. For instance, the Tequendamiense techno-complex was defined on the basis of a few lithic instruments, made on foreign raw materials using supposedly a more elaborate manufacture, but use-wear analyses (Nieuwenhuis, C.J. 2002) indicated that both complexes belong to the same tradition. This also occurs with other lithic techno-complexes used by the author dispersed across South America. I strongly suggest performing a more detailed review of the lithic techno-complexes

included, as some of them are no longer valid according to current techno-morphological criteria.

6) The author mentions that two regions, namely the Orinoco and Amazon basins, are underrepresented in the dataset analyzed. However, I do not see dates from well-known sites in Colombia that belong to such geographical areas, such as Peña Roja and Serrania de la Lindosa. Both sites have extensive 14C dating, so several dates are available. This means that a more comprehensive update about recently published archaeologically-derived 14C dates in South America must be performed to present a more robust analysis.

7) Finally, I suggest including much more detail regarding the Bayesian modeling. Given the use of several parameters to create each chronometric model, it would be important for the reader to clarify the different choices made.

Reviewer #2

(Remarks to the Author)

Dear author,

This study explores the impact of climate on the early human settlement of South America during the late Pleistocene. It investigates how climatic phases, notably the Antarctic Cold Reversal and the Younger Dryas (YD), influenced human migration, adaptation, and megafauna exploitation. The author analysed over 130 archaeological sites and 1,700 dates using a Bayesian modelling approach, and found that human settlement likely started in the southern Andes during or before the Antarctic Cold Reversal. The author concluded that cold conditions were not being a significant barrier of human expansion, while cultural adaptations and milder climate shifts in the Southern Hemisphere benefitted human survival. The study highlights gaps in chronometric data and underrepresentation in certain regions, indicating the need for further research and improved documentation to refine the cultural timeline. It also points out that the link between human activity and megafauna extinction is inconclusive, needing more demographic data to understand extinction patterns.

The idea of this manuscript is original and promising, but while the dating part appears solid, the rest of the manuscript is primarily descriptive and speculative, lacking the formal analytical rigor that I would have expected. The study has potential, but the absence of deeper analysis weakens its impact. In my opinion, significant revisions are necessary to clarify its objectives, expand on methods, and provide a more balanced, comprehensive approach. Below I summarised the main points that I consider as being important to discuss and to justify in a possible revision of the manuscript.

Main comments:

Introduction:

- The introduction lacks sufficient background and is heavily skewed toward archaeology without adequately situating the broader context. Several key concepts, such as the cultural timeline and legacy data concerns, need further explanation to make them clearer to the reader.
- The manuscript seems to present two separate aims, especially with the opportunistic inclusion of the megafauna extinction element, which is underdeveloped. The manuscript needs a clear statement of its primary objective, followed by well-defined, testable hypotheses. These hypotheses should make clear predictions that can be supported or refuted through empirical data and analyses. While the robustness of the dating itself is not in question, the use of contextualisation alone, without more quantitative analysis, is insufficient for drawing robust conclusions given modern analytical tools available.

Methods:

- The rationale for selecting a Bayesian approach over frequentist methods is not fully explained. The advantages of this choice should be briefly discussed.
- The focus on Antarctic cores undermines the argument about ACR's regional impact. A more robust regional climate comparison would enhance the study. You are still presenting the TraCE21k simulations in Supplementary Information but not really using these results to strengthen your conclusion.
- Following up on my former comment, I think that there is a missed opportunity to include more regional climate, vegetation, and population reconstructions. Many existing papers provide formal analyses in this area, and the study relies too much on contextual evidence without sufficient quantification.
- Suggested (non-comprehensive) relevant literature that should be considered to discuss, add formal analyses or comparisons to strengthen the study: (i) Timmermann & Friedrich, *Nature* (2016), (ii) Araujo et al., *Quat. Int.* (2017), (iii) Bartlett et al., *Ecography* (2016), (iv) Sandom et al., *Proc. R. Soc. Lond. B.* (2014), (v) Emery-Wetherell et al., *Paleobiology* (2017), (vi) Saltré et al., *Nat. Commun.* (2019, 2024)

Results & Discussion:

- The proposed timing of human arrival seems early, relying predominantly on literature that supports this view, which introduces a risk of confirmation bias. Studies using frequentist methods, suggesting later arrivals, should be more thoroughly examined and discussed.
- A broader comparison with non-archaeological studies (e.g. including demographic models) is crucial. These results imply earlier human arrivals globally, and (unlike what is argued in the introduction) has implication that cannot be assessed independently of the rest of the world. This broader context needs to be integrated into the discussion to reconcile conflicting

findings from other regions

- Suggested (non-comprehensive) relevant literature: (i) Metcalf et al., *Sci. Adv.* (2016), (ii) Goldberg et al., *Nature* (2016), (iii) Timmermann & Friedrich, *Nature* (2016), (iv) Saltré et al., *Nat. Commun.* (2024)

Minor comments:

- L38-41: The argument that early settlement in South America should be assessed independently from North America is unclear and needs to be better framed. While this makes sense from a climate perspective, the spatial connectivity and demographic links suggest that South America's peopling is directly tied to events in North America unless an alternate migration route was found.
- L54: How are you "correlating" multiple archaeological sites? Several figures present probability density functions in descending order, but there's no apparent quantitative correlation between these outputs.
- L62: I might have missed it but why is there no mention in the main text of the regional climate simulations from TraCE21k, which are shown in the Supplementary Information?
- L62-65: This sentence is overly technical and seems targeted toward archaeologists only. Broader accessibility is needed.
- L107: You assume that your readership is familiar with Bampi et al., but just referencing the paper is insufficient. An overview of the approach would help clarify the context and to follow your logic.
- L159-160: This needs a bit more support. Under this present form, there is no analysis or figure in the main text supporting this conclusion.
- L184-193: This paragraph feels misplaced. Up until this point, the focus has been on the role of ACR in human expansion, but now the topic shifts to megafauna extinction. If both topics are to be included, the framework of the study needs restructuring.
- L206-207: The correlation mentioned here seems speculative. It would benefit from quantification.
- L249: As with Bampi et al., the spatial guidelines by Marsh et al. are mentioned without adequate explanation. Simply referencing the paper is insufficient; an overview of the approach is necessary for clarity.

Figures:

- Including a 3-panel figure (map) in the main text using the first three figures of the Supplementary Information would significantly improve clarity.
- Figure 1: Adding the number of records used for each estimate would be helpful.
- While Figure 3 contains useful information, the results seem peripheral to the paper's main message and would be better suited to the Supplementary Information. Also, please remove "Mosaic plot showing" from the caption.
- Figure 4: These results do not seem to fit the manuscript. If retained, please remove "plot showing..." from the caption.

Supplementary information:

- While there is substantial work and valuable information included in the Supplementary Information, its current format lacks utility. It reads more like a report of exhaustive analyses, listing results site-by-site without summarising or contextualising them to support the main text.
- The code used for plotting figures should be made available on GitHub rather than being included in the Supplementary Information.
- Climate data from the TraCE21k simulations is presented in the Supplementary Information but not explicitly referenced in the main text. This connection should be clarified.

Version 1:

Reviewer comments:

Reviewer #1

(Remarks to the Author)

In the corrected version of the manuscript entitled "Climate Influence on the Early Occupation of South America During the Late Pleistocene," Becerra-Valdivia has clarified several issues, incorporated multiple improvements, and addressed most of the reviewers' suggestions. I am now much more confident in these results and the supporting evidence. I suggest accepting this version with minor revisions, as I believe the author still needs to clarify some points before it can be considered publishable.

Below, I outline these points:

1. I suggest including the word human in the title, as follows: "Climate Influence on the Early Human Occupation of South America During the Late Pleistocene." Adding this term enhances clarity regarding the species under investigation.
2. The claim that the results support the notion that human activity likely began in the southernmost and high-altitude areas introduces potential biases, as earlier dates would typically be expected in northernmost and low-altitude regions. Although the manuscript mentions underrepresentation in certain areas due to research and funding biases, gaps in the archaeological and chronometric records exist across both northern and southern South America related to factors—such as geological, taphonomic, and site formation processes issues—could also contribute to these gaps. Therefore, underrepresentation is not solely attributable to research and funding biases but also to the nature of the archaeological record. This should be explicitly stated in the manuscript to provide a more balanced perspective on the biases discussed.

3. Since the manuscript references recent research suggesting pre-ACR human occupations in certain areas of South America, this study—indicating the earliest occupations in southern South America— and could complement the studies cited in the discussion on line 194 (refs. 34, 37–45):

Del Papa, M., De Los Reyes, M., Poiré, D. G., Rascovan, N., Jofré, G., et al. (2024). Anthropogenic cut marks in extinct megafauna bones from the Pampean region (Argentina) at the last glacial maximum. *PLOS ONE*, 19(7), e0304956. <https://doi.org/10.1371/journal.pone.0304956>

4. The author frequently asserts that “accumulated cultural adaptation” was a key factor in the survival of early human groups during the ACR across South America. However, she does not explain how accumulated cultural knowledge would have facilitated this survival, nor does she specify the mechanisms, pathways, or processes involved. I believe the author should provide explicit scenarios illustrating how accumulated cultural adaptation functioned during this phase of human peopling. This clarification is essential, as adaptation through the accumulation of cultural knowledge and skills is not the only means of adjusting to new or changing environments. Other factors, such as technological innovations, biological adaptations, and cultural or behavioral modifications, may have also played a role. It is important to demonstrate these processes rather than merely assume them.

5. Late Pleistocene hunter-gatherers in South America are generally considered generalists rather than specialists (Gnecco, 2003; Borrero, 2009, 2015). These groups modified their surrounding landscapes as part of a broad range of strategies, not necessarily linked to resource procurement. Given this context, I am curious as to why the author describes landscape modification per se as a specialized activity (line 249). I disagree with this characterization and believe it should be clarified.

6. In line 208, the manuscript states, “this is consistent with numerical models,” but the more appropriate term would be modelling, as this word encompasses the entire process of using numerical models, including problem formulation, model development, validation, and simulation.

Reviewer #2

(Remarks to the Author)

The author has done an excellent job addressing my previous concerns regarding this manuscript. This revised version is much clearer, and I applaud the effort put into improving it. I have no further major comments and am pleased to recommend this manuscript for publication, provided that the minor points below are addressed.

Minor Comments

L38/43: The Bølling-Allerød acronym is written as B-A in L38 but as BA in L43. Please ensure consistency throughout the text.

L64: I assume SH stands for Southern Hemisphere, but it has not been defined. The same applies to NH in L292.

Acronyms: I generally find excessive use of acronyms problematic, as too many different abbreviations can slow down reading and require the reader to repeatedly recall their meanings. While some acronyms are widely recognized and can remain, others (e.g., PDF, SWW, SH) seem unnecessary due to infrequent or inconsistent use (e.g., B-A in L38, BA in L43, but Bølling-Allerød in L252). I suggest minimizing the use of acronyms where possible.

L73: Check if Central Andean Pluvial Event and South American Summer Monsoon System require uppercase consistently throughout the manuscript.

L204: Providing either a brief definition or a reference for the rain shadow effect would be helpful for the reader.

L216-218: The statement “This temporal gap, combined with the inherent limitation of LADs, which record the death of a single individual rather than the point of effective species extinction or decline (the latter likely more informative)” would benefit from a supporting reference, such as:

+ Signor, P. W. & Lipps, J. H. in *Geological implications of large asteroids and comets on the Earth* (Silver, L. T. & Schultz, P. H. eds) (Geological Society of America, 1982).

+ Saltré, F. et al. Climate change not to blame for late Quaternary megafauna extinctions in Australia. *Nat. Commun.* 7, 10511 (2016).

L385: Please remove the parentheses around (PDF).

Reviewer responses (NCOMMS-24-54497)

Reviewer comments are in black, with responses in blue.

Reviewer 1

- 1. The inferred relationship between human activity and paleoclimatic change is ambiguous. A robust way to evaluate the magnitude of such an association is by using the dynamic time warping (DTW) approach, which allows us to measure the similarity (or dissimilarity) between two or more timelines. By using such an approach or a similar one, the author can demonstrate, rather than assume, an association and then defend the role of climate changes in early human peopling dynamics in South America.*

The reviewer correctly notes that no formal statistical correlation was made between the human timeline (represented by multiple probability density functions or PDFs for the onset of cultural occupation across various categories, e.g., provinces) and the ACR/YD phases. This was due to the fact that the PDFs clearly overlap or fall within/outside both ACR and YD at 95% CI (previous Figure 1). However, I now carry out statistical analysis as suggested and outline details in the Methods section. Instead of turning the PDFs into synthetic time series through DTW or similar algorithms, however, I can leave the PDFs as they are and test for correlation within OxCal using the *Difference* command, representing the ACR and YD as uniform distributions 14,500-13,000 and 13,000-11,700 cal BP, respectively. This analysis has been added to the SI (now Supplementary Figure 9). As anticipated, this does not alter the findings.

- 2. The author focused on two climatic phases, ACR and YD. However, regional variations complicate extrapolations to the complete South American subcontinent. For instance, late Pleistocene climatic change in Northern South America and most of Eastern South America was influenced by seasonal variations in wind and rainfall dynamics, that is, the so-called Intertropical Convergence Zone (ICZ). Given its importance for regional human peopling dynamics, at least in Eastern and Northern South America, I recommend including additional proxies from the Cariaco Basin. The author should also consider using other regional proxies like the Sajama Ice Core from Bolivia important to the Central Andean archaeological record.*

I have included more regional climatic data in the introduction. However, because the focus of the paper is indeed the ACR and YD in Southern Hemisphere South America, which excludes northern South America, the focus of the Bayesian modelling discussion is within this region. I now emphasise this in the manuscript.

The Sajama Ice Core is one of the proxies used in the characterisation of the Central Andean Pluvial Event, which was previously noted in lines 179-182 when discussing high-altitude occupation in the Central Andean region.

- 3. In the 14C dataset included as supplementary materials, there are a lot of 14C dates below 7000 14C BP. Were such dates used in the Bayesian modeling for the early stage of human peopling? If so, why? I understood the author assessed cultural phases by using Bayesian*

modeling, but it is not clear to me why such dates are included in a study focused on the late Pleistocene.

In compiling the dataset, I included all available chronometric evidence for each site, even measurements falling outside the primary period of interest (ACR-YR; both younger and older), to ensure comprehensive reporting. For multi-phase models, all archaeo-chronometric data were incorporated, which means each site was modelled in its entirety—this includes layers/phases that, whilst younger/older and beyond the immediate focus of the study, form part of the complete stratigraphic sequence and inform it. I now include a concise version of this response within the Methods.

4. The author mentioned she used ~ 1700 14C dates, however, the criteria used to include or exclude 14C dates are unclear. For instance, a basic issue to exclude a 14C date would be its high error, but in such a dataset there are several 14C dates with a high error, especially those obtained decades ago using beta-counting techniques. The author needs to clarify the quality control procedures followed.

This was explained in previous lines 286-294. In brief, Bayesian modelling can accommodate imprecision and, in many cases, improve it. For instance, a highly imprecise but accurate measurement in layer B will be refined when it is surrounded by well-constrained layers A (above) and C (beneath). Furthermore, as a radiocarbon dating specialist, I approach the manual exclusion of dates with caution, especially when quality control criteria may stem from misunderstandings. In the literature, imprecision is frequently mistaken for inaccuracy, and ages produced by certain pretreatment methods are occasionally discounted due to misperceptions of inefficiency. For these reasons, an objective, transparent, and statistically-informed approach, such as Outlier analysis, is favoured. I now expand on the Methods section to improve this explanation.

5. *I suggest caution with the use of certain not-well-defined lithic techno-complexes in the South American archaeological record. For instance, the Tequendamiense techno-complex was defined on the basis of a few lithic instruments, made on foreign raw materials using supposedly a more elaborate manufacture, but use-wear analyses (Nieuwenhuis, C.J. 2002) indicated that both complexes belong to the same tradition. This also occurs with other lithic techno-complexes used by the author dispersed across South America. I strongly suggest performing a more detailed review of the lithic techno-complexes included, as some of them are no longer valid according to current techno-morphological criteria.*

I appreciate the reviewer's caution regarding the use of certain less well-defined lithic techno-complexes in the South American archaeological record; lithic categorisation can indeed be complex. In this case, however, the Tequendamiense and Abriense complexes are temporally coeval, so whether they are treated as distinct or combined does not affect the results. I have added a note in the Supplementary Information acknowledging Nieuwenhuis' (2002)¹ assessment that they represent a single tradition, as well as Correal Urrego's (2003)² rebuttal. Beyond the Tequendamiense/Abriense complex, the other traditions included—such as El Jobo, Fishtail/Fell, Huentelauquén, Itaparica, Paiján, Pay Paso, and Tigre—are established and valid technological complexes/traditions. I have double checked this with South American lithic experts working in the region.

6. *The author mentions that two regions, namely the Orinoco and Amazon basins, are underrepresented in the dataset analyzed. However, I do not see dates from well-known sites in Colombia that belong to such geographical areas, such as Peña Roja and Serranía de la Lindosa. Both sites have extensive 14C dating, so several dates are available. This means that a more comprehensive update about recently published archaeologically-derived 14C dates in South America must be performed to present a more robust analysis.*

Thank you to the reviewer for highlighting these sites. I have now included the three ACR-YD-aged rock shelters within the Serranía de la Lindosa into the analysis (see SI). Dates for Peña Roja are younger than the period studied at <9250 BP³, so the site has not been included. Whilst it is possible that some data may have been missed, and additional sites are likely to emerge before publication (I am currently dating additional South American sites), the dataset remains broadly representative. The observed patterns—such as the scarcity of data in regions like the Orinoco and Amazon lowlands—likely reflect genuine underlying distributions rather than any unintentional selection bias; the archaeo-chronometric record for these regions during this period is indeed more limited than that of South Hemisphere South America (also shown by Prates et al.⁴ and Goldberg et al.⁵).

7. *Finally, I suggest including much more detail regarding the Bayesian modeling. Given the use of several parameters to create each chronometric model, it would be important for the reader to clarify the different choices made.*

I have expanded on the Methods section. As previously noted, however, the construction of the models followed a set of rules (previous lines 296-303):

In this study, Bayesian model construction for archaeological sites followed a set of rules:

1. *If the site contained multiple stratigraphic levels and measurements, a multi-phase (uniform, as above) model was created.*
2. *If ≥ 2 measurements, the site was represented by a uniform, single-phase model. This is also the case for sites that either lacked sufficient information, e.g., clear stratigraphic descriptions, for multi-phase modelling.*
3. *If represented by only one (non-sediment) measurement dating a single cultural event, no site-specific modelling was done; the single measurement was used.*

Default parameters in OxCal were applied consistently across all models. The General outlier model, for example, was applied in all multi-phase models with a prior probability of 5% using default distribution and scale, with the SSimple outlier model only used for measurement replicates (again, with default distribution and scale). Thus, aside from varying likelihoods (dates) and priors (stratigraphic data) for each site, modelling parameters remained consistent. Sensitivity testing was conducted where likelihoods or priors could be modelled differently to ensure consistency in the posteriors (results). For instance, at Santa Elina, two models were compared—one using original bone apatite measurements and another with calculated bone collagen dates. The results showed no difference in the posterior distributions, affirming the stability of the model outcomes (see SI).

Reviewer 2

Introduction

1. *The introduction lacks sufficient background and is heavily skewed toward archaeology without adequately situating the broader context. Several key concepts, such as the cultural timeline and legacy data concerns, need further explanation to make them clearer to the reader.*

I have edited the introduction and expanded on key concepts.

2. *The manuscript seems to present two separate aims, especially with the opportunistic inclusion of the megafauna extinction element, which is underdeveloped. The manuscript needs a clear statement of its primary objective, followed by well-defined, testable hypotheses. These hypotheses should make clear predictions that can be supported or refuted through empirical data and analyses. While the robustness of the dating itself is not in question, the use of contextualisation alone, without more quantitative analysis, is insufficient for drawing robust conclusions given modern analytical tools available.*

The introduction has been edited, retaining the single objective: ‘to test the role of the ACR and YD on the early settlement of South America’ (previous line 51). In terms of quantitative analysis and in line with this objective, I have now quantified the degree of overlap between the cultural timeline (as separate distributions) and the ACR/YD (Supplementary Figure 9). The conclusions remain the same.

Discussing the results in light of megafaunal extinction evidence is not opportunistic—particularly given the conclusion, which does not jump to point out definitive culprits—but essential. Removing this would be an important omission, particularly as hunting/scavenging evidence is a key category in the archaeological record (like geographic province, lithic technology, etc). This would be possible, however, since the timing of megafaunal extinctions is outside the scope of this paper and represents a separate modelling project. The latter is because, as mentioned in previous lines 187-188 and echoed by others (including suggested literature⁶), last appearance dates do not denote the timing of a species’ effective extinction or population decline (the latter likely more informative). I have expanded on this section in the text.

Methods

3. *The rationale for selecting a Bayesian approach over frequentist methods is not fully explained. The advantages of this choice should be briefly discussed.*

A frequentist approach was (and still is) used (previous Figure 2) and contextualised (previous lines 204-209) within the analysis, in conjunction with the Bayesian approach. I also mentioned that frequentist methods usually applied to archeo-chronometric data, e.g., ⁴C-dates-as-data analysis⁷, are problematic. This argument is now expanded in the Methods, and the overall preference for a Bayesian approach explained.

4. *The focus on Antarctic cores undermines the argument about ACR's regional impact. A more robust regional climate comparison would enhance the study. You are still presenting the TraCE21k simulations in Supplementary Information but not really using these results to strengthen your conclusion. Following up on my former comment, I think that there is a missed opportunity to include more regional climate, vegetation, and population reconstructions. Many existing papers provide formal analyses in this area, and the study relies too much on contextual evidence without sufficient quantification. Suggested (non-comprehensive) relevant literature that should be considered to discuss, add formal analyses or comparisons to strengthen the study: (i) Timmermann & Friedrich, *Nature* (2016), (ii) Araujo et al., *Quat. Int.* (2017), (iii) Bartlett et al., *Ecography* (2016), (iv) Sandom et al., *Proc. R. Soc. Lond. B.* (2014), (v) Emery-Wetherell et al., *Paleobiology* (2017), (vi) Saltré et al., *Nat. Commun.* (2019, 2024)*

The Antarctic cores are critical for defining the timing of the ACR (e.g., Pedro et al.⁸), so it is important to acknowledge their utility whilst clarifying that ice-core and radiocarbon chronologies are not perfectly synchronous (as noted in previous lines 61–65). I have removed the ice-core data from previous Figure 1 to avoid potential confusion and incorporated the explanation from lines 61–65 into the Methods section instead (following the suggestion in point 10). There, I further outline how the degree of overlap between the cultural timeline and the ACR-YD period was tested.

The TraCE21k simulations were referred to in previous lines 164–166 to support the argument that temperature changes during the ACR and Bølling-Allerød (BA) were likely less pronounced in the Southern Hemisphere compared to the Northern Hemisphere.

I recognise the reviewer's point and have now added more regionally specific information to the introduction, as well as expanding the discussion to incorporate further literature and contrast findings.

Results & Discussion

5. *The proposed timing of human arrival seems early, relying predominantly on literature that supports this view, which introduces a risk of confirmation bias. Studies using frequentist methods, suggesting later arrivals, should be more thoroughly examined and discussed.*

I am unsure as to what the reviewer means by suggesting that the paper relies on literature in support of an early (what is considered 'early'?) human arrival, given that the paper deals with a period that *postdates* the earliest human signal in the continent (previous lines 161-164). The paper is not directly dealing with the controversial issue of *when* humans first entered South America (there is convincing evidence of pre-ACR human presence in South⁹ and North America^{10,11}), but rather on how the ACR and YD impacted the settlement process. I have now added a figure (Supplementary Figure 10) in the SI showing pre-ACR cultural evidence to further clarify and reference pre-ACR evidence in Figure 2 (asterisks). Please see point 3 regarding frequentist methods.

6. *A broader comparison with non-archaeological studies (e.g. including demographic models) is crucial. These results imply earlier human arrivals globally, and (unlike what is argued in*

the introduction) has implication that cannot be assess independently of the rest of the world. This broader context needs to be integrated into the discussion to reconcile conflicting findings from other regions. Suggested (non-comprehensive) relevant literature: (i) Metcalf et al., Sci. Adv. (2016), (ii) Goldberg et al., Nature (2016), (iii) Timmermann & Friedrich, Nature (2016), (iv) Saltré et al., Nat. Commun. (2024)

Demography was dealt with in previous lines 206-212, referencing genetics-based evidence. Demographic models based merely on archaeo-chronometric data employing frequentist methods are problematic for many reasons. I expand on this within the text.

As previously mentioned (point 5), this paper does not argue for earlier human arrivals to the region, nor does it suggest that all evidence must be assessed in isolation (next point 7). Regarding the latter, I appreciate the reviewer's suggested literature and have expanded the discussion to frame these results within a broader context, focusing on the adjacent North American region as the most relevant comparison, whilst treating northwestern Europe as a non-analogue due to its distinct dynamics. Expanding the spatial scope further to include other regions, as suggested, risks oversimplifying complex, region-specific dynamics and should be approached with caution, or avoided altogether, when the contrasts in human settlement histories, geography, and climatic contexts are too pronounced.

7. *L38-41: The argument that early settlement in South America should be assessed independently from North America is unclear and needs to be better framed. While this makes sense from a climate perspective, the spatial connectivity and demographic links suggest that South America's peopling is directly tied to events in North America unless an alternate migration route was found.*

I have reframed this as suggested to limit to the climatic perspective, as was intended.

8. *L54: How are you "correlating" multiple archaeological sites? Several figures present probability density functions in descending order, but there's no apparent quantitative correlation between these outputs.*

This was explained in previous lines 305-311. I have edited the Methods section to further clarify.

9. *L62: I might have missed it but why is there no mention in the main text of the regional climate simulations from TraCE21k, which are shown in the Supplementary Information?*

See point 4.

10. *L62-65: This sentence is overly technical and seems targeted toward archaeologists only. Broader accessibility is needed.*

See point 4.

11. L107: *You assume that your readership is familiar with Bampi et al., but just referencing the paper is insufficient. An overview of the approach would help clarify the context and to follow your logic.*

I have now removed Bampi et al.¹² since the reference was unnecessary (I merely cross-checked with this study, finding additional sites since their analysis on or before 2022).

12. L159-160: *This needs a bit more support. Under this present form, there is no analysis or figure in the main text supporting this conclusion.*

This is supported by the results in previous lines 76-80 & 103-105, and depicted in previous Figure 1.

13. L184-193: *This paragraph feels misplaced. Up until this point, the focus has been on the role of ACR in human expansion, but now the topic shifts to megafauna extinction. If both topics are to be included, the framework of the study needs restructuring.*

This section is addressing the results, which include the start of megafaunal hunting/scavenging by humans. In the results (and in previous Figure 1), this follows the reporting of findings according to province, lithic technology and high-altitude evidence (in that order). I continue this sequence when discussing the results (focusing first on the ACR, and then the YD).

14. L206-207: *The correlation mentioned here seems speculative. It would benefit from quantification.*

This is warning against interpreting an increase in archaeo-chronometric data for the period as an increase in human population. I have now reframed to clarify (there is no need for quantification).

15. L249: *As with Bampi et al., the spatial guidelines by Marsh et al. are mentioned without adequate explanation. Simply referencing the paper is insufficient; an overview of the approach is necessary for clarity.*

I have now expanded on this in the Methods.

Figures

16. *Including a 3-panel figure (map) in the main text using the first three figures of the Supplementary Information would significantly improve clarity.*

Thank you for the recommendation. I have now done this, but also added a fourth (presence or absence of megafaunal hunting/scavenging).

17. *Figure 1: Adding the number of records used for each estimate would be helpful.*

Thank you for the recommendation. I have now done this.

18. *While Figure 3 contains useful information, the results seem peripheral to the paper's main message and would be better suited to the Supplementary Information. Also, please remove "Mosaic plot showing" from the caption.*

I have removed the text 'mosaic plot showing', and have now placed it in the SI.

19. *Figure 4: These results do not seem to fit the manuscript. If retained, please remove "plot showing..." from the caption.*

I have removed the text 'plot showing'. The figure, plotting LADs for fauna present in archaeological contexts with direct evidence of killing/scavenging in relation alongside the estimated start for human exploitation, supports the conclusion that 'exploitation of [megafauna by humans] began considerably earlier', depicting this visually.

Supplementary information

20. *While there is substantial work and valuable information included in the Supplementary Information, its current format lacks utility. It reads more like a report of exhaustive analyses, listing results site-by-site without summarising or contextualising them to support the main text.*

While the SI document includes substantial work and valuable information, the reviewer notes that its current format lacks utility, suggesting it reads more like a report of exhaustive analyses rather than summarising or contextualising the results to support the main text. However, I am unclear about the specific concern regarding the lack of summary or contextualisation of individual sites in the SI. The descriptive details for each site are necessary because the Bayesian models rely on stratigraphic information as priors, requiring contextual data to be explicitly outlined. Information such as lithic technology, location, and other site-specific characteristics are essential for categorising sites, which directly informs the large-scale models discussed in the main text. These connections are explicitly addressed through the Figures and OxCal code provided in the 'Supplementary Figure' section. Additionally, each site report in the SI explains the modelling results, including sensitivity testing, which examines whether changes in priors affect the posterior distributions—a critical step in validating the conclusions of any modelling work. The OxCal CQL code is also included after each site to ensure transparency and accessibility for those wishing to replicate or further explore the analysis. The data provided in the SI, whilst detailed, is both statistically summarised and designed to complement the main text, where the broader contextualisation of the findings occurs. As such, the SI is intended to support the main text by offering essential details and methodological transparency.

21. *The code used for plotting figures should be made available on GitHub rather than being included in the Supplementary Information.*

OxCal code is written in its own language, Chronological Query Language (CQL), and its inclusion in GitHub may provide limited additional utility since users would still need to copy it from either the SI or GitHub. However, if the reviewer and Editor prefer, I am happy to comply and upload the OxCal code to GitHub. That said, I believe that readers of this study are likely to focus on specific

site models, making it more practical and useful to feature the CQL code alongside corresponding site reports and figures. This ensures that the context necessary for interpreting the models/code is easily accessible.

22. *Climate data from the TraCE21k simulations is presented in the Supplementary Information but not explicitly referenced in the main text. This connection should be clarified.*

See point 4.

References

1. Nieuwenhuis, C. J. *Traces on tropical tools; a functional study of preceramic sites in Colombia.* (University of Leiden, Netherlands, 2002).
2. Correal Urrego, G. Aclaraciones al texto ‘Traces on tropical tools. A functional study of chert artefacts from preceramic sites in Colombia’ (Nieuwenhuis, Channah José, 2002). *Maguaré* (2003).
3. Arroyo-Kalin, M., Marcote-Ríos, G., Lozada-Mendieta, N. & Veal, L. Entre La Pedrera y Araracuara: la arqueología del medio río Caquetá. *Revista del Museo de La Plata* **4**, 305–330 (2019).
4. Prates, L., Politis, G. G. & Perez, S. I. Rapid radiation of humans in South America after the last glacial maximum: A radiocarbon-based study. *PLoS One* **15**, e0236023 (2020).
5. Goldberg, A., Mychajliw, A. M. & Hadly, E. A. Post-invasion demography of prehistoric humans in South America. *Nature* **532**, 232–235 (2016).
6. Emery-Wetherell, M. M., McHorse, B. K. & Byrd Davis, E. Spatially explicit analysis sheds new light on the Pleistocene megafaunal extinction in North America. *Paleobiology* **43**, 642–655 (2017).
7. Williams, A. N. The use of summed radiocarbon probability distributions in archaeology: a review of methods. *J. Archaeol. Sci.* **39**, 578–589 (2012).
8. Pedro, J. B. *et al.* The spatial extent and dynamics of the Antarctic Cold Reversal. *Nature Geoscience* **9**, 51–55 (2016).
9. Pansani, T. R. *et al.* Evidence of artefacts made of giant sloth bones in central Brazil around

the last glacial maximum. *Proc. Biol. Sci.* **290**, 20230316 (2023).

10. Bennett, M. R. *et al.* Evidence of humans in North America during the Last Glacial Maximum. *Science* **373**, 1528–1531 (2021).
11. Ardelean, C. F. *et al.* Evidence of human occupation in Mexico around the Last Glacial Maximum. *Nature* (2020).
12. Bampi, H., Barberi, M. & Lima-Ribeiro, M. S. Megafauna kill sites in South America: A critical review. *Quat. Sci. Rev.* **298**, 107851 (2022).

Reviewer responses (NCOMMS-24-54497A)

Reviewer comments are in black, with responses in blue.

Reviewer 1

I suggest including the word human in the title, as follows: “Climate Influence on the Early Human Occupation of South America During the Late Pleistocene.” Adding this term enhances clarity regarding the species under investigation.

Good idea. I've added 'human' to the title.

The claim that the results support the notion that human activity likely began in the southernmost and high-altitude areas introduces potential biases, as earlier dates would typically be expected in northernmost and low-altitude regions. Although the manuscript mentions underrepresentation in certain areas due to research and funding biases, gaps in the archaeological and chronometric records exist across both northern and southern South America related to factors—such as geological, taphonomic, and site formation processes issues—could also contribute to these gaps. Therefore, underrepresentation is not solely attributable to research and funding biases but also to the nature of the archaeological record. This should be explicitly stated in the manuscript to provide a more balanced perspective on the biases discussed.

Fair point. Now amended to include:

This underrepresentation likely results from a combination of taphonomic processes, research and funding biases, and poor sample preservation for radiocarbon dating in hot, humid regions such as the Guiana Highlands and the Orinoco and Amazon Lowlands.

Since the manuscript references recent research suggesting pre-ACR human occupations in certain areas of South America, this study—indicating the earliest occupations in southern South America— and could complement the studies cited in the discussion on line 194 (refs. 34, 37–45): Del Papa, M., De Los Reyes, M., Poiré, D. G., Rascovan, N., Jofré, G., et al. (2024). Anthropogenic cut marks in extinct megafauna bones from the Pampean region (Argentina) at the last glacial maximum. PLOS ONE, 19(7), e0304956. <https://doi.org/10.1371/journal.pone.0304956>

I have gone through the suggested literature, but the radiocarbon dates (n = 2) for the reportedly human-modified, faunal specimen are problematic. The first date is on hydroxyapatite from a bone in a wet context, yet there is abundant evidence to suggest that dating hydroxyapatite is only reliable in arid environments (Zazzo and Saliège, 2011). The second date is on a shell found below the reportedly cultural level and there is no consideration of freshwater effects, which would make the age erroneously too old (Philippsen, 2013). The site chronology requires further work, in my opinion. For this reason, I do not include the site as another pre-ACR example in South America.

The author frequently asserts that “accumulated cultural adaptation” was a key factor in the survival of early human groups during the ACR across South America. However, she does not explain how accumulated cultural knowledge would have facilitated this survival, nor does she

specify the mechanisms, pathways, or processes involved. I believe the author should provide explicit scenarios illustrating how accumulated cultural adaptation functioned during this phase of human peopling. This clarification is essential, as adaptation through the accumulation of cultural knowledge and skills is not the only means of adjusting to new or changing environments. Other factors, such as technological innovations, biological adaptations, and cultural or behavioral modifications, may have also played a role. It is important to demonstrate these processes rather than merely assume them.

The term is rooted in the concept advanced by Richerson and Boyd (2005, as cited) and further developed in later work (including a *PNAS* paper now also cited), and refers to the cumulative, intergenerational transmission of knowledge and technologies (i.e., culture) that likely provided ancient humans with an adaptive advantage. This includes technological innovations and cultural/behavioral modifications, as noted by the reviewer. Richerson and Boyd's studies further explore the concept and provide relevant examples. I have now clarified in the text that this is an established concept.

Late Pleistocene hunter-gatherers in South America are generally considered generalists rather than specialists (Gnecco, 2003; Borrero, 2009, 2015). These groups modified their surrounding landscapes as part of a broad range of strategies, not necessarily linked to resource procurement. Given this context, I am curious as to why the author describes landscape modification per se as a specialized activity (line 249). I disagree with this characterization and believe it should be clarified.

I agree that perhaps a poor choice of terminology. I have now removed and left only 'plant domestication', which is undoubtedly a later, Holocene occurrence (cited references).

In line 208, the manuscript states, "this is consistent with numerical models," but the more appropriate term would be modelling, as this word encompasses the entire process of using numerical models, including problem formulation, model development, validation, and simulation.

I added 'numerical models' in response to Reviewer 2's feedback on the previous round. Since 'models' already implies modeling, I am unsure if the change is truly necessary. However, I am happy to comply if the Editor deems it appropriate.

Reviewer 2

L38/43: The Bølling-Allerød acronym is written as B-A in L38 but as BA in L43. Please ensure consistency throughout the text.

Thank you. Done.

L64: I assume SH stands for Southern Hemisphere, but it has not been defined. The same applies to NH in L292.

Thank you. SH defined earlier in the text. I have now defined NH in the first occurrence.

Acronyms: I generally find excessive use of acronyms problematic, as too many different abbreviations can slow down reading and require the reader to repeatedly recall their meanings. While some acronyms are widely recognized and can remain, others (e.g., PDF, SWW, SH) seem unnecessary due to infrequent or inconsistent use (e.g., B-A in L38, BA in L43, but Bølling-Allerød in L252). I suggest minimizing the use of acronyms where possible.

Thank you. I have corrected and tried to reduce the acronyms.

L73: Check if Central Andean Pluvial Event and South American Summer Monsoon System require uppercase consistently throughout the manuscript.

Thank you. I have checked.

L204: Providing either a brief definition or a reference for the rain shadow effect would be helpful for the reader.

Thank you. I actually provided a reference, which I have now moved to the right place so the reader can refer.

L216-218: The statement “This temporal gap, combined with the inherent limitation of LADs, which record the death of a single individual rather than the point of effective species extinction or decline (the latter likely more informative)” would benefit from a supporting reference, such as:

+ Signor, P. W. & Lipps, J. H. in Geological implications of large asteroids and comets on the Earth (Silver, L. T. & Schultz, P. H. eds) (Geological Society of America, 1982).

+ Saltré, F. et al. Climate change not to blame for late Quaternary megafauna extinctions in Australia. Nat. Commun. 7, 10511 (2016).

I've gone through the suggested references but neither is entirely applicable and so I have not included. The first one is an edited volume centred on asteroid impacts and the second one is on megafaunal extinctions in Australia.

L385: Please remove the parentheses around (PDF).

I have checked and there is no PDF in line 385.

References

Philippsen B (2013) The freshwater reservoir effect in radiocarbon dating. *Heritage Science* 1(1): 24.

Zazzo A and Saliège J-F (2011) Radiocarbon dating of biological apatites: A review. *Palaeogeography, palaeoclimatology, palaeoecology* 310(1): 52–61.

In the corrected version of the manuscript entitled “*Climate Influence on the Early Occupation of South America During the Late Pleistocene*,” Becerra-Valdivia has clarified several issues, incorporated multiple improvements, and addressed most of the reviewers’ suggestions. I am now much more confident in these results and the supporting evidence. I suggest accepting this version with minor revisions, as I believe the author still needs to clarify some points before it can be considered publishable.

Below, I outline these points:

1. I suggest including the word *human* in the title, as follows: “*Climate Influence on the Early Human Occupation of South America During the Late Pleistocene*.” Adding this term enhances clarity regarding the species under investigation.
2. The claim that the results support the notion that human activity likely began in the southernmost and high-altitude areas introduces potential biases, as earlier dates would typically be expected in northernmost and low-altitude regions. Although the manuscript mentions underrepresentation in certain areas due to research and funding biases, gaps in the archaeological and chronometric records exist across both northern and southern South America related to factors—such as geological, taphonomic, and site formation processes issues—could also contribute to these gaps. Therefore, underrepresentation is not solely attributable to research and funding biases but also to the nature of the archaeological record. This should be explicitly stated in the manuscript to provide a more balanced perspective on the biases discussed.
3. Since the manuscript references recent research suggesting pre-ACR human occupations in certain areas of South America, this study—indicating the earliest occupations in southern South America— and could complement the studies cited in the discussion on line 194 (refs. 34, 37–45):

Del Papa, M., De Los Reyes, M., Poiré, D. G., Rascovan, N., Jofré, G., et al. (2024). *Anthropic cut marks in extinct megafauna bones from the Pampean region (Argentina) at the last glacial maximum*. PLOS ONE, 19(7), e0304956. <https://doi.org/10.1371/journal.pone.0304956>

4. The author frequently asserts that “*accumulated cultural adaptation*” was a key factor in the survival of early human groups during the ACR across South America. However, she does not explain how accumulated cultural knowledge would have facilitated this survival, nor does she specify the mechanisms, pathways, or processes involved. I believe the author should provide explicit scenarios illustrating how accumulated cultural adaptation functioned during this phase of human peopling. This clarification is essential, as adaptation through the accumulation of cultural knowledge and skills is not the only means of adjusting to new or changing environments. Other factors, such as technological innovations, biological adaptations, and cultural or behavioral modifications, may have also played a role. It is important to demonstrate these processes rather than merely assume them.
5. Late Pleistocene hunter-gatherers in South America are generally considered generalists rather than specialists (Gnecco, 2003; Borrero, 2009, 2015). These groups modified their surrounding landscapes as part of a broad range of strategies, not necessarily linked to resource procurement. Given this context, I am curious as to why the author describes landscape modification per se as a specialized activity (line 249). I disagree with this characterization and believe it should be clarified.
6. In line 208, the manuscript states, “*this is consistent with numerical models*,” but the more appropriate term would be *modelling*, as this word encompasses the entire process

of using numerical models, including problem formulation, model development, validation, and simulation.